# Effects of Hybrid Polymeric Material Based on Polycaprolactone on the Environment

**DOI:** 10.3390/ma15144868

**Published:** 2022-07-13

**Authors:** Maria E. Fortună, Elena Ungureanu, Doina C. Jităreanu, Denis C. Țopa, Valeria Harabagiu

**Affiliations:** 1Institute of Macromolecular Chemistry “Petru Poni”, 41A Grigore Ghica Voda Alley, 700487 Iasi, Romania; fortuna.maria@icmpp.ro (M.E.F.); hvaleria@icmpp.ro (V.H.); 2Department of Plant Science, “Ion Ionescu de la Brad” University of Life Sciences, 3 Mihail Sadoveanu Alley, 700490 Iasi, Romania; doinaj@uaiasi.ro (D.C.J.); dennistopa@uaiasi.ro (D.C.Ț.)

**Keywords:** aminopropyl-terminated polydimethylsiloxane, environmental impact, ε-caprolactone, hybrid, tomato plants (*Lypercosium esculentum*), microorganisms

## Abstract

Polymers are of great interest in areas such as agriculture, medicine and pharmacy, the food and cosmetic industries, and the chemical and construction industries. However, many polymers are nonbiodegradable and are not environmentally friendly. They are highly resistant to degradation and therefore can lead to waste disposal problems. In recent years, the interest in the microbial degradation of polymeric materials has grown due to the desire for less waste pollution in the environment. In this study, the biodegradable polymer that was obtained by the ring-opening polymerization of ε-caprolactone (CL) using an aminopropyl-polydimethylsiloxane (APDMS) oligomer and the effects of the polymer towards the growth and development of tomato plants (*Lypercosium esculentum*) were investigated. The obtained product was characterized using FTIR spectroscopy, NMR spectroscopy, and energy dispersion spectroscopy (EDX) analysis, and the effects of this compound on the evolution of tomato plants (*Lypercosium esculentum*) were studied. We also studied the biological stability of the product by identifying some of the microorganisms that developed on the surface, given its susceptibility to biodegradation.

## 1. Introduction

Due to their mechanical and chemical properties with numerous applications, polymers play an important role in a multitude of fields [1]. In general, synthetic polymers offer excellent mechanical and thermal properties, while biobased polymers are mostly biodegradable. Therefore, by combining the advantages of natural polymers and synthetic polymers, the resulting hybrid materials have the potential for biomedical and environmental applications [2].

The polymers with potential toxicity that accumulate in nature influence the environment and the various organisms exposed to them. For this reason, the environmental impacts of polymers and their waste should be studied [3].

A series of polymers are nonbiodegradable and are not decomposed by microorganisms, and this can lead to waste disposal problems; therefore, an alternative is needed for the disintegration of this waste. The fabrication of materials containing both biodegradable and nonbiodegradable polymers can be an effective method to reduce the total amount of polymer waste, which cannot be degraded in nature [4].

Biodegradation is a very valued process that harnesses the power of the microorganisms present in an environment to remove plastic products in a safe and efficient way [5]. The microorganisms growing in the rhizosphere have a significant role in the biodegradation of the hybrid materials [6,7]. The composites suffer different changes by being decomposed into simple organic compounds that modify the metabolism of the plant, leading to the modification of the plant structure. The quantity of nitrogen from the roots and stalks of the plant varies both according to the type of product used and its addition [8].

Contact between the microorganisms and the hybrid structure will result in an increase in the contact surface between the microorganisms and the synthetic polymer and acceleration of biotic reactions [9]. Other organisms, such as insects, snails, or beetles, can also contribute to the degradation process by digesting these products, and through biodegradation, they can form ketone groups, thus accelerating the destruction of the synthetic matrix [2]. Therefore, biodegradation is influenced by many factors: the nature and structure of the polymer and its compounds, reaction conditions, and microorganisms [10,11]. Given these aspects, natural or synthetic and degradable or non-degradable polymers have received much more attention in recent decades [12,13].

Polydimethylsiloxane (PDMS) is a synthetic polymer with many interesting properties such as chemical stability, biocompatibility, a low glass transition temperature (T_g_), gas permeability, high hydrophobicity, and resistance to biodegradation [14]. Therefore, it can be used for functionalizing various surfaces and as a building block for polymers and hybrid organic–inorganic systems [15,16,17]. Due to its properties, PDMS has a wide range of applications, and its elimination has an impact on the environment. PDMS is nonbiodegradable and is missing in the environmental impact studies of siloxanes and the siloxanes compounds [18]. However, it is assumed that the siloxane stability in biodegradation is due to the lowered electron density in the carbon atoms and in the polydimethylsiloxanes and a steric shielding effect in the methyl groups. The insertion of functional groups into polysiloxane chains may be a promising approach to benign organosilicon compounds [19,20,21].

Polycaprolactone (PCL) is a semi-crystalline polymer with native biocompatibility and biodegradability, a high solubility at room temperature, and a low melting temperature and is known to be miscible with a large variety of polymers. PCL degradation is readily degraded by the lipases and esterase of the microorganism [22].

The block copolymers are obtained through a combination of excellent properties of PDMS and PCL, which makes it a synthetic biomaterial with various applications, particularly biomedical ones. In this regard, the literature has reported it as a method for the preparation of block copolymers and for the ring-opening polymerization of ε-caprolactone with mono or difunctionally terminated reactive hydroxy or amine functional oligomers as initiators [23,24].

In this study, the synthesis and characterization of the block copolymer of polycaprolactone with aminopropyl-terminated polydimethylsiloxane and the effects of this compound on the growth and development of tomato plants (*Lypercosium esculentum*) along with the isolation and characterization of microorganisms involved in environmental impact were evaluated comparatively with the starting materials.

## 2. Experimental Section

### 2.1. Materials

Black peat that was harvested in September 2020 was offered by “Ion Ionescu de la Brad” Iasi University of Life Sciences (IULS) and was used as a soil with the following properties: 58.14% relative humidity; 41.82% dry substance; 3.84% ash; 96.16% volatile substances; 76.00% porosity; 38.12% water retention capacity; pH = 7.12; C% = 52.29; N_t_% = 0.94; C/N = 55.62; P(P_2_O_5_)% = 0.14; K(K_2_O)% = 0.12. Peat is predominantly found in hill and mountain regions and consists of a set of organic matter (decomposed leaves, grass, fungi, or insects) and has a number of advantages over ordinary soil: it accelerates seed germination and plant growth; allows for higher yields, both quantitatively and qualitatively, in terms of economic efficiency; it has a neutral or slightly alkaline pH; and has a higher mineral content [2].

Biological material: Tomato seeds (*Lypercosium esculentum*), San Marzano variety, IULS. The fruits have an elongated form and the weight is 60/80 g.

Chemical materials: ε-Caprolactone and octamethylcyclotetrasiloxane (D4) were purchased from Aldrich; stannous octoate was obtained from Air Products. Toluene was purchased from Aldrich and was freshly used after distillation over sodium wires. Xylene and methanol were purchased from Sigma-Aldrich (St. Louis, MO, USA).

### 2.2. Synthesis of Aminopropyl-Terminated Polydimethylsiloxane

APDMS was synthesized using the chemical equilibration of 1,3-Bis(3-aminopropyl)-1,1,3,3-tetramethyldisiloxane with octamethylcyclotetrasiloxane in the presence of tetramethylammonium hydroxide catalyst, according to a previous experiment [25].

In a round-bottom flask, a solution of tetramethylammonium hydroxide (10%) in methanol was prepared under magnetic stirring. Then, 0.8 g of D4 and 10 mL of toluene were added. The reaction mixture was stirred for 12 h to 80 °C under an argon atmosphere. APDMS was obtained after the distillation of octamethylcyclotetrasiloxane at 150 °C under vacuum (1 mmHg). The solvent was also removed under vacuum.

### 2.3. Synthesis of Aminopropyl-Terminated Polydimethylsiloxane-Caprolactone (APDMS-CL) Copolymers

APDMS-CL copolymer was obtained using ring-opening polymerization of ε-Caprolactone in the presence of stannous octoate as the catalyst (Figure 1) [26,27].

In a reaction, 3 g CL, 1.6 g APDMS as a macroinitiator, 0.023 g of stannous octoate catalyst, and 2 g of xylene were added to a 50 mL three-neck round bottom flask. Then, the reaction mixture was heated to 120 °C and stirred under nitrogen for 24 h. Due to viscosity, 10 g of xylene was added and stirred. Purification was performed by precipitating the reaction mixture in methanol. The precipitate was dried in a vacuum oven at 40 °C for 24 h, and the product was accessed to have a yield of 91.5%.

### 2.4. Characterization Methods

Fourier Transform Infrared (FTIR) spectroscopy was used to investigate the surface chemistry of the synthesized materials. The absorption spectra were recorded using a Bruker Vertex 70 FTIR spectrometer (Bruker Optics, Ettlingen, Germany). Registrations were performed in transmission mode within a 400–4000 cm^−1^ range with a resolution of 2 cm^−1^ at room temperature on samples dissolved in KBr pellets. Nuclear Magnetic Resonance (^1^H-NMR) spectra were recorded with a Bruker NMR spectrometer at 400 MHz (Bruker, Billerica, MA, USA). The standard for a chemical shift in CDCl_3_ (deuterate chloroform) is TMS (dilute tetramethylsilane).

The surface morphologies were visualized by Scanning Electron Microscopy (SEM) (Brno, Czech Republic) and a Quanta 200 scanning electron microscope (5 kV) with an EDX elemental analysis system (Ametek, Berwyn, PA, USA) that was used to identify the surface morphology and composition.

Thermogravimetric (TGA) and differential scanning calorimetry (DSC) analyses were performed using STA 449F1 Jupiter equipment (Netzsch, Selb, Germany) and a DSC 200 F3 Maia device (Netzsch, Selb, Germany), respectively.

The environmental impact of the siloxane matrices was carried out by evaluating parameters that provide clues on the initiation of the ecotoxicological impact, such as capacity and germination index, plant size, amount of green and dry biomass, nitrogen dynamics, and the study of biological stability was determined by identifying biological agents in the system.

The experimental laboratory conditions consisted of 23–25 °C for 35 days, during which time visual observations and biometric analyses were carried out. Vegetation vessels (height of 9 cm and diameter of 6 cm) and a culture medium consisting of black peat were used as supports [3]. The sample weights of APDMS, CL, and APDMS-CL were 0.3 g.

The mass of the soil introduced into each vessel was 80 g. Firstly, a layer of soil (3 cm) was placed into each vessel, then the fraction of the sample (APDMS, CL, and APDMS-CL) was added. A 4 cm layer of soil was added on top of that, and then 3 tomato seeds were added, which were covered with a 2 cm layer of soil. As a reference, the soil was only used with tomato seeds (Reference sample). The experimental samples are notated as follows: Reference, CL, APDMS, and APDMS-CL.

Subsequently, the non-disintegrated solid siloxane fractions were collected and transported in sterile bags for mycological testing.

In order to correctly determine any installed microorganisms, the samples were inserted into Petri plates on wet filter paper and submitted to thermostatation. After that, the samples were passed on to a culture medium containing 40 g NaNO_3_ (sodium nitrate), 20 g K_2_HPO_4_ (dipotassium hydrogen phosphate), 10 g MgSO_4_ (magnesium sulphate), 10 g KCl (potassium chloride), 0.2 g FeSO_4_ (iron sulphate), 30 g sucrose, and 20 g agar. Finally, the samples were dissolved in 1000 mL distilled water, in which micromycetes needed to develop in order to correctly determine the genus and species concerned. The resulting medium was sterilized for 20 min at 120 °C and then distributed in Petri dishes with diameters of 10 cm, followed by another sterilization. The micromycetes that appeared after 48 h on the matrix fragments were repainted. These were cultivated until the appearance of fungal colonies. All cultures were incubated in a dark cultivation chamber for 28 days at a temperature of 25 ± 2 °C and a relative air humidity of 80 ± 2% [28]. At the end of the test period, macro- and microscopic examination was performed to detect and identify any microorganisms.

For the microscopic analysis, the classical method was used using the Kern OBN 135 microscope, RS 136-3498. For greater accuracy of the results, mass spectrometry was used—a high-performance method that precisely identifies the unique “molecular footprint” of each microorganism, which was achieved by using a Maldi TOF MS spectrophotometer [29,30,31].

The IG germination index was then calculated using the following Equation (1):(1)$$\mathrm{IG}=x\;100\frac{\mathrm{number}\;\mathrm{of}\;\mathrm{germinated}\;\mathrm{seeds}\;\left(\%\right)\;x\;\mathrm{root}\;\mathrm{length}\;\mathrm{for}\;\mathrm{samples}\;}{\;\mathrm{number}\;\mathrm{of}\;\mathrm{germinated}\;\mathrm{seeds}\;\left(\%\right)\;x\;\mathrm{root}\;\mathrm{length}\;\mathrm{for}\;\mathrm{reference}\;\mathrm{sample}}\;(\%)$$

## 3. Results and Discussion

### 3.1. Structural Characterization

The samples were investigated by Fourier Transform Infrared spectroscopy. The results confirm the synthesis of APDMS-CL triblock copolymers (Figure 1).

The APDMS structure was confirmed by the presence of a –CH_2_–NH_2_ bend at about 1588 cm^−1^, which was assigned to the amine group. The 1023–1093 cm^−1^ bands were assigned to Si–O–Si, and the 800 cm^−1^ band was attributed to CH_2_–Si–O vibrations. The 1261 cm^−1^ band was assigned to Si–CH_3_. The APDMS-CL structure was confirmed by the presence of the absorption band at 1099 cm^−1^, which was assignable to the stretching vibration of the Si–O bonds of the APDMS polymer, and the presence of the very strong stretching vibration of the carbonyl groups at 1728 cm^−1^ was attributed to the ester group in CL. The 1018–1099 cm^−1^ region was assigned to the Si–O–Si bands, the band at 1261 cm^−1^ was attributed to the Si–CH_3_ band, and 800 cm^−1^ was attributed to the CH_2_–Si–O vibrations.

The amide I (from the C=O stretching vibrations) and amide II (from the C–N stretching and CNH) bands correspond to 1651 and 1546 cm^−1^, respectively [32,33].

The synthesis of the well-defined copolymer was confirmed by the NMR spectrum (Figure 2).

The attribution of the peaks of the copolymer, as seen in Figure 2, were as follows: –CH_2_–H (3.61 ppm); –CH_2_–NH– (4.04 ppm); –CH_2_–CH_2_–CH_2_–NH– (1.62 ppm); –CO–CH_2_–CH_2_–CH_2_– (1.42 ppm); –CO–CH_2_–CH_2_– (1.62 ppm); –CO–CH_2_– (2.32 ppm); –Si(CH_3_)_2_– (0.07 ppm); –Si–CH_2_–(0.5 ppm); and –CH_2_–CH_2_–CH_2_ (1.47).

### 3.2. Surface Morphology

An EDX elemental analysis system was used to identify the surface morphology and composition. The elemental analysis was calculated after three repeated measurements.

Figure 3 illustrates the EDX spectra for the APDMS-CL copolymer. The presence of siloxane was confirmed by the high peak at around 1.8 keV, which is characteristic of silicon atoms.

A scanning electron microscope was used to study the phase separation. The SEM results indicate that the APDMS-CL sample has a phase segregation morphology (Figure 4).

### 3.3. Thermal Properties of the APDMS and APDMS-CL

Shown in Figure 5 are the registered TG and differential thermogravimetric (DTG) curves of APDMS (a) and APDMS-CL (b) under a nitrogen atmosphere. The thermal behavior of these materials was assessed in terms of percentage loss in weight from ambient temperature up to 700 °C.

More weight loss steps could be observed for the APDMS sample, with maximum decomposition peaks situated at about: 80, 140, 260, 376, 408, 451, and 550 °C. It could be observed that APDMS has four weight loss steps that occur at 146.65 °C (with the weight loss of 5.58%), at 422.97 °C (with a weight loss of 11.70%), at 473.15 °C (with a weight loss of 16.11%), and at 587.28 °C (with a weight loss of 19.75%), and at 699.48 °C, a residual mass of 2.48% remains. The weight loss in the range of the residual mass corresponds to material decomposition and the leftover mass residue. For the APDMS-CL composite, two maximum decomposition peaks situated at 359 and 443 °C could be observed. It could also be observed that APDMS-CL has two weight loss steps that occur at 373.59 °C (with the weight loss of 91.14%), and at 479.74 °C (with a weight loss of 3.17%), and at 699.47 °C, a residual mass of 5.06% remains.

The DSC analysis for APDMS and its APDMS-CL composite were made in reference to temperature values between −100 and 200 °C (Figure 6).

The first measurements (first heating) showed a glass transition (T_g_) at –114 °C (two peaks centered at about 46 and 90 °C) for APDMS and at –122 °C for the APDMS-CL composite (a peak at about 62 °C). For the second measurement (second heating), the glass transition was obtained at –117 °C and –122 °C for APDMS and APDMA-CL, respectively. As heating continued, a shift was observed in the composites yielding peaks at 24 °C, potentially indicating crystallization.

In this study, biometric testing analyzes the evolution of tomato plants, and microbiological analysis assesses the biological stability of the samples.

### 3.4. Tomato Seed Germination Test

The germination test is the most sensitive parameter for the assessment of toxicity and a method of estimating the nutritional qualities of the soil, i.e., the substrate obtained when applied as a nutrient to plant growth. Seed germination is influenced by temperature, with each plant having a minimum threshold, an optimal threshold, and a maximum threshold of the thermal values between which this process takes place.

During the growing season, all of the experimental samples were visually monitored periodically, every 7 days, and biometric measurements were taken for the purpose of assessing the development of plant size (Figure 7).

In order to determine the germination index, it was necessary to measure the length of the plant root. Table 1 presents the average length of the plant roots 35 days after planting and the germination index.

From the experimental data analysis, the size of the roots varied in descending order, as follows: Reference—1.6 cm, CL—1.4 cm, APDMS—1.1 cm, and APDMS-CL—0.7 cm.

As expected, the germination index has the maximum value in the case of the reference sample, a fairly high value, approximately 90%, in the CL sample, a value of more than 50% in the APDMS sample, and a value slightly below 50% in the APDMS-CL sample.

### 3.5. The Average Height of the Plants

The average height of the tomato plants for every sample is presented in Figure 8.

The plant height for APDMS had smaller development than what was recorded for the reference, CL, and APDMS-CL samples and degraded more slowly. It seems that the state of liquid aggregation allows for faster interaction with the soil, and the effect on the growth and development of the plants is a more pronounced one.

In the case of APDMS-CL, after 14 days, the size of the plant had similar development to the reference sample, which can be explained by the non-initiation of biodegradation. After 21, 28, and 35 days, the size of the plant is less evolutionary for APDMS-CL and CL than in the reference sample, which confirms the initiation of biodegradation, but is higher than in APDMS, which had faster interaction with the soil, due to its liquid state.

In the CL sample, the size of the plant evolved similar to that of the reference sample, with no major variations.

Plant growth and development are below the reference level for all of the samples studied, except for the CL sample, which behaves similarly to the reference, but there is no major decrease, which can be attributed to the susceptibility of the samples to the environment to some extent.

### 3.6. Dry and Green Biomass

Subsequently, in order to determine the amount of green biomass, the tomato plants were weighed on the analytical balance. Because the plants were smaller in size, the weighing was conducted in its entirety. After three days of weighing the green biomass, dry biomass weighing was also carried out (the drying of the plants was carried out under laboratory conditions) (Figure 9).

Figure 10 represents the dry tomato plants for all of the samples three days after harvest.

From the experimental data analysis, the amount of green biomass varies in descending order, as follows: Reference, CL, APDMS, and APDMS-CL. Additionally, the quantity of dry biomass varies in the same order. Compared to the reference sample, the values obtained for the samples with siloxane content show that it influences the green biomass and the dry biomass, which demonstrates the initiation of the ecotoxicological impact of these matrices and the less disturbing effect on plant evolution. In order to determine the germination index, it was necessary to measure the length of the plant root. Table 2 presents the total nitrogen determination after plant removal.

After analyzing the values of this parameter, it can be concluded that the matrices incorporated in the soil are susceptible to ecotoxicological impact and influence the growth and development of plants, showing an involution compared to the reference, but not on a major scale.

### 3.7. Total Soil Nitrogen Dynamics

The next stage of the study consisted of the determination of the total nitrogen using the Kjeldhal method for all of the soil samples, which constituted the culture medium for the tomato seeds. Nitrogen fixation is vital for plants, although very few species (soybeans, clover, peas, beans, lentils) have the ability to fix nitrogen from the atmosphere through bacteria.

The nitrogen deficiency in tomatoes is manifested by the yellowing of the leaves from the tip to the petiole along the main rib, and in the case of seedlings, the plants are light green, which are characteristics that were not visible in our experiment. The nitrogen amount in the soil depends on two antagonistic processes: mineralization and immobilization.

The first (mineralization) consists of the decomposition of the organic substance under the action of soil microorganisms and the release of ammonium ions, and the second (immobilization) consists of the conversion of mineral nitrogen from the soil, also under the action of microorganisms, into cellular proteins and other compounds with organic nitrogen [34].
(2)$$N\;\mathrm{from}\;\mathrm{the}\;\mathrm{soil}\overset{\mathrm{mineralization}}{\underset{\mathrm{immobilization}}{⇌}}{\mathrm{NH}}_{4}{}^{+}\overset{\mathrm{nitrification}}{\underset{\begin{array}{l} \mathrm{nitrifying}\;\mathrm{reduction}\\ \mathrm{and}\;\mathrm{immobilization} \end{array}}{⇌}}{\mathrm{NO}}_{3}{}^{-}$$

Regarding the dynamics of the total nitrogen in the soil, the differences between the samples studied are small and statistically insignificant.

It is known that most plants need nitrogen from the soil and for most crops; this means that they are dependent on nitrogen fertilizers and beneficial bacteria from the soil [35]. This is also confirmed by the experimental data obtained, as the nitrogen content of the reference sample in the final measurement is lower than that recorded in the reference sample in the initial measurement.

In comparison, it was found that the lowest nitrogen content is recorded in the reference sample, while in the other experimental samples, it is higher, which demonstrates the contribution of organic nitrogen released from the studied samples.

The visual analysis during the growing season of the experimental samples, the biometric measurements, and the amounts of green and dry biomass conclude that the plants had enough nitrogen for normal growth and development.

### 3.8. Biological Stability

The impact of the siloxane matrices as well as their susceptibility to the environment were also tested by a biological stability study based on the isolation and characterization of the microorganisms involved.

On the one hand, the presence of siloxane can act selectively by inhibiting the action of microorganisms that would biodegrade the matrix, while on the other hand, it allows for the selection of species that are tolerant to these polymeric structures.

At the end of the experiment, the visual analysis found that in the case of the APDMS-CL sample, there were traces of undisintegrated compounds, so the siloxane matrices were harvested and subsequently transported in sterile bags for mycological testing. In the other samples studied, mycological testing could not be performed.

Two species of fungi (Fusidium viride and Penicillium brevi-compactum) were identified from the analyses, with colony diameters varying inversely proportionally to the amount of solid siloxane matrix (the larger the diameter, the smaller the amount of siloxane) (Figure 11). The macroscopic observations [9,36] on the degree of the development of fungal mycelium on APDMS-CL are presented in Table 3.

The isolated Fusidium viride fungus formed on Czapek’s agar (CZA) or Czapek–Dox medium and took the form of a very fine filament from which the conidiophores could hardly be distinguished. The oval conidia that may or may not be sharp towards the ends of the filament are hyaline and measure 5–7 × 3 μm (Figure 12).

In *Penicillium brevicompactum*, the colonies that appeared on the medium are quite small and green–gray against the yellow-gray-colored medium under the colony. The diameter of the colony reached 2–3 cm 10–12 days after emergence. The conidiophores varied in length and were shorter than in other Penicillium species, forming areas on the surface of the colony with a higher center and slightly yellow droplets. Under the colony, the mycelium was yellow. The conidiophores were 300 × 3.5 μm, with sparse branches and smooth walls. The branches were 20 × 3 μm, the meticulous were 12 × 2.5 μm, and the fields were 10 × 2.5 μm, and there was an absence of smooth conidia, with ovate to subglobulation measurements of 2.5 × 2.5–3 μm.

The fact that the siloxane matrices function as new hosts for the two micromycetes: Fusidium viride and *Penicillium brevicompactum*, is a first, as this aspect has not been reported in the literature thus far.

Microscopic aspects are edifying for the effects of the studied matrices and for the development of microorganisms [37]. Thus, it is possible on the one hand to highlight the characteristics of fungi capable of biosynthesizing the enzymes that can degrade the siloxane material. On the other hand, these results allow the clarification of the ability to inhibit the development of microorganisms. Plant cultivation also involves the manifestation of the rhizosphere and the stimulation of the development of microorganisms that may be involved in the environmental impact.

## 4. Conclusions

In the light of the parameters studied, the experimental results demonstrated the environmental impact of the tested samples. The products were released into the soil, which influenced the growth and development of the plants. The analyzed parameters showed lower values compared to the reference sample (in the vessel without siloxane material, the plants developed better than if they were introduced into the soil alongside the tested matrices) but were at an acceptable level. The development of microorganisms, in particular, Fusidium viride and *Penicillium brevicompactum*, were first identified on hybrid polymeric materials based on polycaprolactone, which also confirms that these products do not have a major disturbing effect on soil composition and plant evolution and show acceptable susceptibility to the environment.

Moreover, due to the superior performance compared to the individual materials and through the possibilities of modeling the properties according to the field of use, the researched hybrid can be a solution for the future, without negatively influencing the bioremediation of the soil.

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
