# Peer review of "Effects of Hybrid Polymeric Material Based on Polycaprolactone on the Environment"

_materials, 2022, doi:10.3390/ma15144868_

Round 1

Reviewer 1 Report

In this manuscript, the authors synthesized and characterized the APDMS-CL triblock copolymers and investigated the effects of this polymer on the growth and development of tomato plants (Lypercosium esculentum) along with the isolation and characterization of microorganisms. This research is interesting and meaningful, some comments should be considered as follow:

  1. The content is broken into too many paragraphs in the introduction. In lines 75-77, the authors mentioned “The block copolymers obtained by a combination of excellent properties of PDMS with the PCL makes these a synthetic biomaterial with various applications, especially, biomedical”, some related reports should be cited and introduced.
  2. In “2.2. Synthesis of aminopropyl-terminated polydimethylsiloxane”, after the solvent was removed under vacuum (10 mm Hg), no further steps were conducted to remove the unreacted octamethylcyclotetrasiloxane?
  3. In 2.3, what is the yield of APDMS-CL copolymer?
  4. For comparison, the NMR spectrum of APDMS, CL should be added to Figure 2.
  5. The TEM analysis of APDMS-CL should be conducted.
  6. More than just presenting the experimental results, more discussion of Figures 5 and 6 should be added.
  7.  The error bars in figures 8-9 should be added, the sample size of the tomato seed germination test is better be increased.
  8. The scale bar of figures 10-11 should be added.
  9. In lines 322-333, “In table 2 are presented the average length of plant root after 35 days of planting and germination index”, but table 2 showed the total nitrogen determination after plant removal, not the average length of plant root.

Author Response

Response to comments of Reviewer 1:

We would like to thank the reviewer for all the valuable comments and remarks which helps us to really improve the scientific quality of our revised manuscript. We have carefully considered all the reviewer comments and have revised the manuscript in light of them. The suggested modifications were clearly marked in red in the revised manuscript.

Reviewer 2 Report

The research presents the synthesis and characterization of the polycaprolactone block copolymer with aminopropyl-terminated polydimethylsiloxane and the influence of this compound on the growth and development of tomato plants, along with the isolation and characterization of microorganisms that should be involved in biodegradation.

The authors claim that they are investigating the biodegradation of the siloxane matrix: "The biodegradation of the siloxane matrices was carried out by evaluating parameters that provide clues on the initiation of the biodegradation process, such as capacity and germination index, plant size, amount of green and dry biomass, nitrogen dynamics and the study of biological stability by identifying biological agents in the system"

But these parameters determine the ecotoxicological impact not necessarily on the bio/degradation products but may also on leached products from the matrix.

There is biodegradation of a polymer when exposed to microorganisms if they cause changes in the chemical structure of a polymer resulting in the loss of a specific property of that polymer.

Please describe material analyzes showing the progress of biodegradation. What parameters prove the biodegradation of the aminopropyl-terminated polydimethylsiloxane block? After degradation, SEM of the surface, DSC and TGA should be taken.

The results were presented in the form of a report. There is a lack of discussion of results, especially in the case of DSC and germination test. Giving a value is not a discussion. The discussion should be linked to some literature references. What is the mass fraction of the PCL block in the copolymer?

Authors should present analyzes proving biodegradation or change the research goal to the ecotoxicological impact of the synthesized material.

Also some additional missing points:

1.       Line 60, 232-237; Variables (e.g. Tg) should be italicized (here also g in subscript) Please correct throughout the text.

2.       Line 78, 467; “e-caprolactone”: Please use Greek letters to denote substituent positions. Please correct throughout the text.

3.       “Lypercosium esculentum” or “Penicillium brevicompactum”; The scientific names of species are italicized. Please correct throughout the text.

4.       " 48 hours”, “3g”; In general, units should be written according to the SI rules: appropriate abbreviations (h not hour), spacing between units and the variable, etc. Please correct throughout the text.

Author Response

Response to comments of Reviewer 2:

We would like to thank the reviewer for all the valuable comments and remarks which helps us to really improve the scientific quality of our revised manuscript. We have carefully considered all the reviewer comments and have revised the manuscript in light of them. The suggested modifications were clearly marked in green in the revised manuscript.

Thank you very much!

The authors

Reviewer 3 Report

Manuscript "Hybrid polymeric material based on polycaprolactone suscepti- 2 ble to biodegradation" is well conceptualized. Overall, results are interesting and it should be considered for publication but only after major revision. Language and style revision is necessary, but my biggest concern is lack of discussion in regards to previous literature findings. Please find my comments and suggestions attached. 

Author Response

Response to comments of Reviewer 3:

We would like to thank the reviewer for all the valuable comments and remarks which helps us to really improve the scientific quality of our revised manuscript. We have carefully considered all the reviewer comments and have revised the manuscript in light of them. The suggested modifications were clearly marked in blue in the revised manuscript.

Thank you very much!

The authors

Round 2

Reviewer 1 Report

The revised manuscript is improved and could be accepted for publication.

Reviewer 2 Report

No additional comments.

Reviewer 3 Report

Authors accepted and followed all the instructions and suggestions.